# Impacts of Energy Structure on Carbon Emissions in China, 1997–2019

**DOI:** 10.3390/ijerph19105850

**Published:** 2022-05-11

**Authors:** Fengjian Ge, Jiangfeng Li, Yi Zhang, Shipeng Ye, Peng Han

**Affiliations:** 1Department of Land Resource Management, School of Public Administration, China University of Geosciences, Wuhan 430074, China; fjge@cug.edu.cn (F.G.); imshipeng@yeah.net (S.Y.); hp980801@cug.edu.cn (P.H.); 2Department of Economics and Management, China University of Geosciences, Wuhan 430074, China; yizhangcug@163.com

**Keywords:** carbon emissions, energy structure, Moran’s *I*, SEMLD, provincial level, China

## Abstract

To mitigate climate change, reducing carbon dioxide (CO_2_) emissions is of paramount importance. China, the largest global CO_2_ emitter, proposes to peak carbon emissions by 2030 and become carbon neutral by 2060; transforming the energy structure represents one of the primary means of addressing carbon emissions; thus, it is essential to investigate the impacts of alternate energy sources throughout the country. Based on energy consumption and carbon emissions data from 30 provincial-level administrative regions in China (excluding Tibet, Hong Kong, Taiwan, and Macau, due to the lack of data), the study here investigated the shares of coal, petroleum, natural gas, and non-fossil energy sources (i.e., hydropower, nuclear power, wind power, solar power, and biomass power), as they relate to total, per capita, and per unit GDP CO_2_ emissions via spatial regression. The results showed that: (1) The epicenters of coal and carbon emissions have shifted from the east to the central and western regions; (2) There is a significant correlation between energy structure and carbon emissions: coal has a positive effect, petroleum’s effects are positive at first, and negative subsequently; while both natural gas and non-fossil energy sources have a negative impact; (3) Provincial-level carbon emissions are affected by energy structure, carbon emissions in neighboring regions, and other factors.

## 1. Introduction

According to the International Energy Agency, China is the world’s largest emitter of greenhouse gases (GHGs) accounting for 28.93% of global emissions, 94.60% of which was in the form of CO_2_ from fossil fuel combustion in 2019, according to the China Statistical Yearbook and the China Energy Statistical Yearbook. Indeed, China’s CO_2_ emissions per unit gross domestic product (GDP) exhibited a continuous downward trend between 1980 and 2019, falling from 5.23 to 1.08 tC· CNY 10,000^−1^ (Chinese Yuan—CNY); although, total CO_2_ emissions and per capita rates increased over the same period, rising from 1.38 to 9.58 billion tC, and from 1.39 to 6.79 tC·persosn^−1^, respectively (Figure 1). Despite these trends, China’s energy structure improved consistently from 1997 to 2019, as the share of fossil fuels dropped from 96.00% to 84.70%, chiefly due to the decreasing share of coal. Over the same period, the share of petroleum remained relatively unchanged, that of natural gas slowly grew, and the shares of non-fossil fuel energy sources (i.e., hydropower, nuclear power, solar power, and wind power) also grew consistently from 4.00% to 15.30%, equating to an increasing proportion of the country’s power generation structure from 19.36% to 31.15% (Figure 2).

Moreover, these proportional shifts were accompanied by variable effects on the CO_2_ emission indicators. Hence, there remains a need to assess the impacts of energy structure on total, per capita, and per unit GDP CO_2_ emissions, thereby informing future carbon reduction policies in China for combatting global climate change.

While studies regarding the impacts of individual energy components on carbon emissions remain limited, there has been an abundance of research on factors influencing carbon emissions, the majority of which indicating that energy structure is one of the most important factors influencing carbon emissions. The main research methods used in such studies were factor decomposition and econometric analyses, where the former tends to use the logarithmic mean Divisia index (LMDI) for studying the impacts of dynamic changes in various factors, such as energy consumption intensity and structure, as well as industry structure, GDP, and population, on carbon emissions [1,2,3,4,5,6,7,8,9,10,11,12]. For example, Bhattacharyya et al. studied 15 European Union member nations using LMDI-based factor decomposition, finding that changes in energy structure, reduced energy intensity, and improved production processes were the primary drivers of carbon emissions [7]. Alternatively, econometric analyses typically involve the use of time series, cross-sectional, and panel data models to study the correlations of technological progress, population size, urbanization, and other factors with carbon emissions [13,14,15,16,17,18,19,20]. Moreover, the geographic detector method (i.e., geodetector), has become a dominant technique for investigating factors influencing the spatial distribution patterns of carbon emissions since being first proposed by Wang et al. [21]. Additionally, spatial regression models have also been applied to the spatiotemporal patterning of carbon emissions given their intuitive, rapid, and comprehensive nature.

In China, several studies have focused on the effects of population growth, industries, urbanization, energy structure, and energy consumption intensity on carbon emissions. Du constructed a panel data model of provincial-level carbon emissions, finding that the shares of coal, heavy industry, as well as urbanization levels, have a significant positive correlation with carbon emissions in China, whereas the relationship between per capita CO_2_ emissions and the level of economic development displayed a significantly inverted U-shape [22]. Both Yang and Liu, and Wan et al. analyzed factors influencing regional differences in carbon emissions across China using the stochastic impacts by regression on population, affluence, and technology (STIRPAT) model, concluding that energy and industry structure, energy intensity, per capita GDP, and urban population size play a critical role in carbon emissions based on per capita energy sources [23,24]. Alternatively, Cao and Zeng and Wang and He analyzed the driving factors behind regional differences in carbon emissions across the Yangtze River Economic Belt, and those influencing provincial-level CO_2_ emissions throughout China, respectively [25,26]. Other scholars have also conducted a series of quantitative studies to reveal the influential factors of energy structure on carbon emissions [27,28,29,30,31,32,33,34].

Accordingly, it has been revealed that population growth, industries, and urbanization affect carbon emissions through energy structure and consumption. Since total energy consumption is primarily influenced by various aspects, including both economic and technological factors, the sector cannot directly dictate total energy consumption, but only manage the supply structure. Accordingly, as carbon reduction mechanisms are explored from the perspective of the energy sector, the research here has analyzed panel data on provincial-level energy structure and carbon emissions in China using a geographically weighted regression model. The purpose of the present study was twofold: (1) To analyze the characteristics of energy structure and CO_2_ emissions at the provincial level in China, and (2) To investigate the impact of energy structure on carbon emissions.

## 2. Materials and Methods

### 2.1. Data Sources and Processing

Since only the data of China’s provincial carbon emissions from 1997 to 2019 can be collected at present, all data used in this study are from that period, a total of 23 years. Here, population and GDP data for the 23-year period since 1997 were sourced from the China Statistical Yearbook (1997–2019), while total energy consumption and energy structure data were obtained from the China Energy Statistical Yearbook (1997–2019). Carbon emissions data were obtained from the China Emission Accounts and Datasets (1997–2019), whereas CO_2_ emissions coefficients were acquired from the 2019 IPCC Guidelines for National Greenhouse Gas Inventories. These datasets were processed using ArcGIS *v*.10.6 and GeoDa *v*.1.16.

The first step in data processing was standardizing the measurement units of energy sources. Here, various types of energy sources were converted to the heat unit measurement of tons of coal equivalent (tce), where one tce is equal to 29,307.6 ∗ 10^3^ kJ or 7000 kcal (according to the International Steam Table calorie). Appendix A presents the conversion factors for various types of energy sources from which the uniform standard heat for each energy source type was obtained.

Second, the total energy consumption from coal, petroleum, natural gas, and non-fossil fuel energy sources was calculated. Total coal consumption was taken as the sum of consumption for various coal product types, including: (1) raw coal, (2) clean coal, (3) other washed coal, (4) briquettes, (5) coke, (6) coke oven gas, (7) other gasses, (8) other coking products, and (18) heat; whereas total petroleum consumption assessed: (9) crude oil, (10) gasoline, (11) kerosene, (12) diesel oil, (13) fuel oil, (14) liquefied petroleum gas—LPG, (15) refinery gas, and (16) other petroleum products. Natural gas consumption was obtained directly from the relevant data on (17) natural gas. All non-fossil fuel energy sources—hydro-, nuclear, wind, and solar power—consisted of two parts: (18) electricity and (19) other energy. Based on provincial-level statistical yearbooks published throughout China, hydro- and nuclear power consumption were included under (18) electricity, while wind, solar, and biomass consumption were placed within (19) other energy. Notably, electricity consumption, represented here solely by hydro- and nuclear power consumption, excluded fossil-fuel power consumption. Since CO_2_ emissions coefficients of non-fossil energy sources were zero, their correlated impacts on emissions were considered negligible. Furthermore, the consumption of non-fossil fuel energy sources was typically presented as a single total in statistical yearbooks, and could not be further partitioned out into individual sources; thus, the analyses of non-fossil energy sources here were combined into a single category.

### 2.2. Methods

#### 2.2.1. Energy Structure and Carbon Emissions

The formula for energy structure was expressed via Equation (1):(1)$$SE_{i}=\frac{E_{i}*ECC{\;}_{i}}{\sum^{\;}E_{i}*ECC{\;}_{i}}$$
where SE represents the energy structure, E is the physical energy consumption, ECC is standard conversion factor for energy sources based on the tce (details are provided in Appendix A), and i is the source energy type.

The sourced carbon emissions data from the China Emission Accounts and Datasets were calculated based on energy consumption, and thus served as the CO_2_ emissions coefficients used for calculation [35,36,37,38].

#### 2.2.2. Exploratory Spatial Data Analysis

The study here assessed the spatial correlations and variations between units of analysis from both global and local perspectives, commonly represented by Moran’s *I* indicator [39]. Global Moran’s *I* was used for detecting spatial correlations across China, to identify overall spatial clustering and characteristics of carbon emissions (total, per capita, and per unit GDP CO_2_ emissions). The formula for global Moran’s *I* is shown in Equation (2):(2)$$\mathrm{Moran}’s\;I=\frac{\sum_{i,j}W_{ij}\left(CE_{i}-\bar{\mathrm{CE}}\right)\left(CE_{j}-\bar{\mathrm{CE}}\right)}{\sigma^{2}\sum_{i,j}W_{ij}}$$
where i and j represent two different provincial-level administrative regions; CE is the CO_2_ emissions; $\bar{CE}$ is the average CO_2_ emissions of the 30 provincial-level administrative regions; $\sigma^{2}$ is the variance; and W is the spatial weight matrix whose determinant is 1 if i and j are adjacent to each other, otherwise it is 0. Global Moran’s *I* range between −1 and 1, where a positive (negative) value represents a positive (negative) correlation, and the greater the absolute value, the stronger the spatial correlation. A global Moran’s *I* of zero indicates no spatial correlation.

As local spatial autocorrelation can be used to identify clusters of similar values among different provinces, it addresses the shortcoming that global spatial autocorrelation cannot reflect spatial clustering characteristics within regions. Accordingly, the formula for local Moran’s *I* is expressed via Equation (3):(3)$$I_{i}=\frac{Y_{i}-\bar{Y}}{\sigma^{2}}\overset{N}{\underset{j=1,j\neq i}{\sum }}W_{ij}\left(Y_{i}-\bar{Y}\right)$$
where $Y_{i}$ and $Y_{j}$ represent CO_2_ emissions in provinces i and j, respectively; $W_{ij}$ represents the row-normalized spatial weight matrix (calculated based on Equation (2)), constructed according to the adjacency and distance rules; and $Y_{i}-\bar{Y}$ represents the standard deviation for the potential of each region.

#### 2.2.3. Spatial Regression Analysis

Ordinary least squares models (OLS, Equation (4)) have been adopted by previous studies since the spatial effects of dependent variables are generally considered independent and identically distributed; however, this can lead to the incomplete interpretation of regression results, as it neglects potential spatial effects [40]. Here, three spatial regression models with different application directions were employed for analysis, including the spatial lag model (SLM, Equation (5)), spatial error model (SEM, Equation (6)), and the spatial error model with lag dependence (SEMLD, Equation (7)) [41,42]. SLM assumes that spatial correlations are present in dependent variables, emphasizing neighborhood effects, and taking into consideration the spatial diffusion (spillover effects) of dependent variables across geographical units. Alternatively, SEM focuses on neglected and unobserved spatial correlations between variables, whereas SEMLD is an extended model of SEM including the addition of spatial lag variables. Lastly, the study here determined the fit of these models using the log-likelihood ratio (LogL), Akaike information criteria (AIC), and the Schwartz criterion (SC), where the greater the LogL and the smaller the AIC or SC values, the better the fit [43]. All models are simulated on GeoDa *v*.1.16.

OLS, SLM, SEM, and SEMLD were expressed via Equations (4)–(7), respectively:(4)$$Y_{t}=\alpha X_{t}+\varepsilon$$
(5)$$Y_{t}=\alpha X_{t}+\beta_{1}W_{1}Y_{t}+\gamma$$
(6)$$Y_{t}=\alpha X_{t}+\beta_{2}W_{2}\varepsilon +\gamma$$
(7)$$Y_{t}=\alpha X_{t}+\beta_{1}W_{1}Y_{t}+\beta_{2}W_{2}\varepsilon +\gamma$$
where $Y_{t}$ represents the matrix of the dependent variable (carbon emissions) at time t; $X_{t}$ represents the matrix of the independent variable (energy structure) at time t; α is the coefficient vector of $X_{t}$, indicating the impact of the independent variable on the dependent variable; $\beta_{1}$ is the spatial lag term; $\beta_{2}$ is the spatial error term; $W_{1}$ and $W_{2}$ represent the spatial weight matrices of the lag and error terms, respectively (calculated based on Equation (2)); ε is the vector of the random error term in the least-squares model; and γ represents the constant error term.

## 3. Results

### 3.1. Spatiotemporal Characteristics of Energy Structure in China

Since only data on carbon emissions from 1997 to 2019 can be collected, to demonstrate the spatiotemporal characteristics of China’s energy structure more clearly from 1997 to 2019, a visual analysis of the energy structure data was conducted, and the results were illustrated across five-time sections: 2000, 2005, 2010, 2015, and 2019. The results of these analyses are detailed as follows:

#### 3.1.1. Coal Consumption

Figure 3 illustrates the spatial shares of coal in China’s provincial-level energy structure. Over the past 20 years, the share of coal increased at first, and declined thereafter. Moreover, trends of coal consumption varied by province. In 2000, Inner Mongolia and Shanxi had the highest shares of coal, followed by Jilin, Hebei, Henan, Anhui, Jiangsu, Jiangxi, Yunnan, and Guizhou. In 2005, the share of coal in Guizhou increased significantly, while declines were recorded in Ningxia and Guangdong. In 2010, the share of coal in Anhui rose markedly, followed by that of Gansu; whereas in 2015, the provinces with the highest coal shares were Inner Mongolia, Ningxia, Gansu, Shanxi, and Shandong, compared to Qinghai, Sichuan, Chongqing, Guangxi, Guangdong, Fujian, Zhejiang, and Beijing with the lowest shares. In 2019, the highest shares of coal were observed in Inner Mongolia and Shanxi, while the lowest were found in Qinghai and Sichuan.

#### 3.1.2. Petroleum Consumption

Figure 4 illustrates the provincial-level proportion of petroleum consumption in China’s energy structure. Since 2000, petroleum’s share has increased, and then declined, subsequently, resulting in relatively minor changes overall. Moreover, petroleum consumption varied by province: in 2000, Guangdong and Hainan demonstrated the highest share of petroleum, followed by Heilongjiang, Fujian, Zhejiang, Beijing, and Tianjin. In 2005, the share of petroleum in Xinjiang, Inner Mongolia, and Chongqing exhibited an upward trend; whereas the remaining provinces showed either a decline or no significant change. In 2010, petroleum’s share in Xinjiang, Ningxia, Shandong, Jiangsu, Guizhou, and Guangdong decreased; however, Gansu, Shaanxi, Yunnan, and Hainan saw an increase, while the remaining provinces showed no significant changes. In 2015, the provinces with the highest petroleum shares were Beijing and Shanghai, followed by Tianjin, Liaoning, Guangdong, and Fujian. In 2019, the highest shares of petroleum were found in Beijing and Shanghai, while the lowest were recorded in Inner Mongolia, Shaanxi, Shanxi, Hebei, Shandong, and Hainan.

#### 3.1.3. Natural Gas Consumption

Figure 5 illustrates the provincial-level natural gas consumption in China’s energy structure. Notably, the share of natural gas has grown over the analysis period. In 2000, Xinjiang, Sichuan, and Chongqing maintained the highest share of natural gas; whereas in 2005, Hainan demonstrated the greatest proportion, followed by Qinghai, with the shares in all remaining provinces remaining low. In 2010, Hainan showed a decline in the natural gas share, while Qinghai and Beijing exhibited an increasing trend, and no significant changes were observed in the remaining provinces. In 2015, Hainan, Tianjin, Beijing, Qinghai, and Gansu experienced an increase in the share of natural gas; whereas in 2019, the provinces with the highest shares were Beijing and Hainan, followed by Tianjin, Qinghai, Sichuan, and Chongqing, with those in all remaining provinces being below the medium level.

#### 3.1.4. Non-Fossil Fuel Energy Consumption

Figure 6 illustrates provincial-level non-fossil fuel energy consumption in China, showing that it has grown markedly since 2000, with the most significant increase observed between 2015 and 2019. In 2000, the highest share of non-fossil fuel energy sources was observed in Ningxia, followed by Qinghai, Yunnan, and Guangxi. In 2005, Ningxia experienced a significant decline in the share of non-fossil fuel energy, while the lowest shares were recorded in Inner Mongolia and Shanxi. In 2010, there were no significant changes in the spatial patterning of non-fossil fuel energy consumption were recorded in any of the provinces; however, Ningxia, Anhui, and Shaanxi exhibited a decline, while an increase was observed in Guizhou, Hunan, and Jiangxi. In 2015, the share of non-fossil energy in Yunnan and Guangxi grew, while in 2019, the highest shares of non-fossil fuel energy sources were Beijing, Sichuan, Yunnan, Guangxi, and Zhejiang, with the lowest in Inner Mongolia, Shanxi, Qinghai, and Anhui.

### 3.2. Spatiotemporal Characteristics of Carbon Emissions in China

To more clearly demonstrate the spatiotemporal characteristics of total, per capita, and per unit GDP CO_2_ emissions in China between 1997 and 2019, this study conducted further visual analyses of the data, and illustrations are provided across five years: 2000, 2005, 2010, 2015, and 2019. The results of these analyses are detailed as follows:

#### 3.2.1. Total CO_2_ Emissions

Total CO_2_ emissions in China have been increasing consistently since 2000, as the number of provinces with >2 million tCO_2_ emissions has risen from only four in 2000, to 27 in 2019, thereby encompassing most of the provinces in China. The center of peak total CO_2_ emissions has gradually shifted from the eastern region (Bohai Rim) to the central and western regions, while the province with the highest total CO_2_ emissions has changed from Liaoning to Shanxi over the past two decades. Regarding the pace of change, total CO_2_ emissions rose at their fastest between 2000 and 2010, but slowed significantly around 2019 (Figure 7).

#### 3.2.2. Per Capita CO_2_ Emissions

Figure 8 illustrates the provincial-level per capita CO_2_ emissions in China, revealing similar upward trends in total CO_2_ emissions since 2000. The distribution of maximum per capita CO_2_ emissions has gradually shifted from the east (Bohai Rim) to the central and western regions. Specifically, the province with the highest per capita emissions shifted from Liaoning to Shanxi over the last two decades. Regarding the pace of change, per capita CO_2_ emissions increased the fastest between 2000 and 2010, slowing gradually thereafter (Figure 8).

#### 3.2.3. CO_2_ Emissions per Unit GDP

Figure 9 illustrates the provincial-level CO_2_ emissions per unit GDP across China, revealing a continuous decline over the past 20 years, with only Shanxi in the central region experiencing an increase over the analysis period. Peak CO_2_ emissions per unit of GDP shifted from the northeast to the central region, while the province with the highest CO_2_ emissions per unit of GDP changed from Inner Mongolia to Shanxi. Regarding the pace of change, emissions per unit GDP declined relatively rapidly between 2000 and 2015, and continued to do so more gradually thereafter (Figure 9).

### 3.3. Spatial Autocorrelation

Spatial correlation analyses were conducted with three dependent variables—total, per capita, and per unit GDP CO_2_ emissions—to obtain the values of global Moran’s *I* between 1997 and 2019 (Table 1). In these analyses, *p* represented the concomitant probability, where *p* < significance level indicated that the value of Moran’s *I* passed the significance test. The values of global Moran’s *I* for per capita CO_2_ emissions fell within the α = 0.05 significance level; thus, per capita emissions in China exhibited a relatively stable spatial clustering pattern, and with significant autocorrelation among China’s 30 provinces.

#### 3.3.1. Total CO_2_ Emissions

According to the local indicators of spatial association (LISA) map for the spatiotemporal distribution of total CO_2_ emissions (Figure 10), the high-high cluster in China exhibited regional clustering, with similar total CO_2_ emissions in 2000, 2005, 2010, 2015, and 2019. In 2000, the high-high cluster primarily included four provinces—Jilin, Hebei, Shandong, and Henan—and was distributed in strips along the northeast coast. Elsewhere, Gansu became part of the low-low cluster, which was located in the central and western regions; whereas neither low-high nor high-low clusters existed. In 2005, some changes were observed to this spatial divergence, as the high-high cluster primarily consisted of five provinces, including Hebei, Shandong, Henan, Shaanxi, and Anhui, while low-low clusters were comprised of two provinces—Xinjiang and Sichuan; however, as with 2000, no low-high or high-low clusters were observed. In 2010, high-high clusters showed no further expansion, although Sichuan became part of the high-low cluster. Alternatively, no changes were recorded in the low-low cluster either, while no low-high clusters were recorded. In 2015, the high-high cluster began to shrink to four primary provinces—Hebei, Shandong, Henan, and Shaanxi; whereas Xinjiang became part of the high-low cluster. Notably, both the low-low and low-high clusters were absent. In 2019, the high-high cluster continued to shrink down to three main provinces—Hebei, Shandong, and Henan—whereas the high-low cluster remained unchanged. Moreover, Sichuan became part of the low-low cluster, while the low-high cluster remained absent.

#### 3.3.2. Per Capita CO_2_ Emissions

According to the LISA map for the spatiotemporal distribution of per capita CO_2_ emissions (Figure 11), the high-high cluster in China exhibited regional spatial clustering, with similar per capita CO_2_ emissions in 2005, 2010, 2015, and 2019. In 2000, the high-high cluster primarily included three provinces—Jilin, Liaoning, and Henan—mainly distributed in strips along the northeast coast; however, no low-high, high-low, or low-low clusters were present. In 2005, there were some significant changes to this spatial divergence, as the high-high cluster expanded to include Inner Mongolia; whereas the low-low cluster mainly consisted of seven provinces—Hubei, Hunan, Jiangxi, Guangdong, Guangxi, Guizhou, and Yunnan—while low-high and high-low clusters were not present. In 2010, the high-high cluster shrank to three provinces—Inner Mongolia, Hebei, and Shaanxi—and the low-low cluster also decreased in size to four provinces—Hunan, Jiangxi, Guangdong, and Guangxi. Alternatively, Guizhou became part of the high-low cluster, while Gansu served as the epicenter of the low-high cluster. In 2015, there were no significant changes to the spatial distribution of the four clusters, as all observations remained relatively consistent with those of 2010. In 2019, the high-high cluster continued to decrease in size to only incorporate Inner Mongolia, while the low-low cluster also shrank to three provinces—Hunan, Guangdong, and Guangxi. The low-high cluster expanded to two provinces—Hebei and Jilin—whereas the high-low cluster remained unchanged.

#### 3.3.3. CO_2_ Emissions per Unit GDP

According to the LISA map for the spatiotemporal distribution of CO_2_ emissions per unit GDP (Figure 12), the high-high cluster in China exhibited regional spatial clustering of similar values in 2005, 2010, 2015, and 2019. In 2000, the high-high cluster primarily included three provinces in the northeast, while the low-high cluster was comprised of two provinces—Ningxia and Sichuan. Neither high-low nor low-low clusters were present. In 2005, there were some changes to this spatial divergence, as the high-high cluster consisted of four provinces in northern China—Inner Mongolia, Gansu, Ningxia, and Shaanxi— while the low-low cluster was now comprised of four provinces along the southeastern coast—Guangdong, Fujian, Guangxi, and Zhejiang. Low-high clusters had disappeared, while high-low clusters did not exist. These findings were consistent with those observed in both 2005 and 2010. In 2015, the high-high cluster expanded to include Shanxi, while the low-low cluster shrank to only include Jiangxi. Alternatively, Henan became part of the low-high cluster, while no high-low clusters were observed. In 2019, the high-high cluster continued to expand, now including Jilin, as did the low-low cluster to incorporate Zhejiang as well; however, the low-high cluster had disappeared, while the high-low cluster remained non-existent.

### 3.4. Impact of Energy Structure on Carbon Emissions in China

#### 3.4.1. Energy Structure on Total CO_2_

When investigating the impacts of energy structure on total CO_2_ emissions, the results of the OLS residual test showed that the values of Moran’s *I* for 2000, 2005, 2010, 2015, and 2019, were −0.861, 0.379, 0.424, 0.454, and 0.378, respectively, suggesting that there was strong spatial autocorrelation among the residuals in this model, while statistically significant spatial lag and error terms were also present. Accordingly, the use of SLM, SEM, and SEMLD in the present study had the potential to improve the goodness of fit. When comparing the LogL, AIC, and SC values of these three models, it was found that the SEMLD model had the greatest LogL, and smallest AIC and SC values, thus indicating its superior explanatory power (Table 2).

The spatial regression results obtained using the SEMLD model relating to the share of coal in 2000, 2005, 2010, 2015, and 2019 (the same order of years applies hereinafter) were 0.523, 0.850, 0.652, 1.284, and 0.723, respectively, notably all of which were positive (Table 3). Based on this trend, the share of coal was found to decrease at first, before increasing and declining again thereafter; thus, there was a positive correlation between the share of coal and total CO_2_ emissions, but the intensity of its impact fluctuated with time. Alternatively, the regression coefficients for petroleum turned from positive to negative: 0.537, 0.239, 0.160, −0.114, and −0.288, respectively, showing that an increase in petroleum’s share led to a rise in total CO_2_ emissions up to 2013, and a decline in total CO_2_ emissions thereafter. Alternatively, the regression coefficients of the share of natural gas were −0.085, −0.065, −0.074, −0.126, and −0.190, respectively, all of which indicated a negative correlation between the share of natural gas and total CO_2_ emissions. Moreover, the absolute values of these coefficients increased with time, revealing that the increasing share of natural gas was correlated with a greater decline in total CO_2_ emissions over time. Lastly, the regression coefficients of the share of non-fossil fuel energy sources were all negative: −0.619, −0.500, −0.019, −0.011, and −0.196. Additionally, the absolute values of these coefficients exhibited a decreasing trend at first, and increased thereafter, confirming that an increase in the share of non-fossil fuel energy sources led to a decline in total CO_2_ emissions; moreover, its impact declined at first, and increased subsequently.

The spatial lag terms of the SEMLD in 2000, 2005, 2010, 2015, and 2019 were significant, indicating that the spatial spillover effect of total CO_2_ emissions in each province of China was significant. On average, for every 1% increase in total CO_2_ emissions within a given province, the total correlated CO_2_ emissions increased by 0.506%, 0.628%, 0.744%, 0.776%, and 0.504%, respectively. Accordingly, such growth in carbon emissions did not come from growth within the province, but rather was imported from neighboring provinces. Thus, carbon emissions in a particular province are not only highly correlated to its own internal factors, but closely related to neighboring factors (e.g., carbon emissions in neighboring provinces) as well. Further, the spatial error terms in all five-time sections were significant at the *α* = 0.01 level, demonstrating that total CO_2_ emissions in China were influenced by coal, petroleum, natural gas, and non-fossil fuel energy sources, in addition to other factors.

#### 3.4.2. Energy Structure on per Capita CO_2_

When investigating the impacts of energy structure on per capita CO_2_ emissions, the results of the OLS residual test showed that the values of Moran’s *I* for the five years of 2000, 2005, 2010, 2015, and 2019 were 3.653, 1.560, 1.725, 2.261, and 0.435, respectively, with *p*-values of 2000, 2010, and 2015 being significant at the *α* = 0.10 level. These findings suggest a strong spatial autocorrelation among the residuals in this model, and confirm the presence of statistically significant spatial lag and error terms as well. When comparing the LogL, AIC, and SC values of the SLM, SEM, and SEMLD models, it was found that the SEMLD model had the greatest LogL value, and the smallest AIC and SC values, again indicating its stronger explanatory power (Table 4).

According to these spatial regression results from the SEMLD model (Table 5), the regression coefficients for the shares of coal in 2000, 2005, 2010, 2015, and 2019 were 0.811, 0.712, 0.656, 0.856, and 0.110, respectively, all of which were positive, and exhibited a decreasing trend with time. These findings confirm a positive correlation between the share of coal and per capita CO_2_ emissions; although, the strength of its impact decreased gradually with time. Alternatively, the regression coefficients for the shares of petroleum shifted from positive to negative over time: 0.630, 0.367. −0.569, −0.058, and −0.619, respectively. These findings suggest that the increase in petroleum’s share led to a rise in per capita CO_2_ emissions up to 2008, and a decline thereafter. For natural gas, the regression coefficients were all negative: −0.428, −0.348, −0.530, −0.623, and −0.757, respectively; however, the absolute values of these coefficients exhibited an increasing overall trend. Thus, the share of natural gas had a negative correlation with CO_2_ emissions per capita, where a continuous increase in the share of natural gas led to a decline in per capita CO_2_ emissions, with the strength of its impact increasing with time. Lastly, the regression coefficients for the shares of non-fossil fuel energy sources were all negative as well: −0.619, −1.035, −0.926, −0.760, and −0.798, respectively; however, the absolute values of these coefficients exhibited a downward trend over the analysis period. Thus, an increase in the share of non-fossil fuel energy sources led to a decline in per capita CO_2_ emissions; although, its impact gradually weakened with time.

The spatial lag terms in the SEMLD model throughout the study period were all significant, indicating the importance of the spatial spillover effect on per capita CO_2_ emissions in each province. The spatial lag terms in 2000, 2005, 2010, 2015, and 2019 were 0.651, 0.693, 0.681, 0.377, and 0.251, respectively, all of which were significant at the *α* = 0.10 level. On average, for every 1% increase in per capita CO_2_ emissions within a particular province in 2000, 2005, 2010, 2015, and 2019, per capita CO_2_ emissions increased by 0.651%, 0.693%, 0.681%, 0.377%, and 0.251%, respectively. In addition, the spatial error terms in 2000, 2005, and 2010 were all significant, showing that China’s per capita CO_2_ emissions in these years were influenced not only by coal, petroleum, natural gas, and non-fossil fuel energy resources, but by other factors as well.

#### 3.4.3. Energy Structure on CO_2_ Emissions per Unit GDP

The OLS residual test results on the impacts of energy structure for CO_2_ emissions per unit GDP showed that the values of Moran’s *I* in 2000, 2005, 2010, 2015, and 2019 were 0.234, 2.068, 3.266, 2.915, and 4.009, respectively. Furthermore, the *p*-values in 2005, 2010, 2015, and 2019 were significant at the *α* = 0.05 level, indicating the presence of strong spatial autocorrelation among the model residuals. When comparing the LogL, AIC, and SC values of the SLM, SEM, and SEMLD, it was found that the SEMLD had the greatest LogL, and the smallest AIC and SC values, thus revealing its stronger explanatory ability (Table 6).

The spatial regression results obtained using the SEMLD model relating to the share of coal in 2000, 2005, 2010, 2015, and 2019 (the same order of years applies hereinafter) were all positive and exhibited a decreasing trend: 1.181, 1.175, 1.133, 1.129, and 0.607, respectively (Table 7). Thus, it was indicated that an increase in coal’s share led to a rise in CO_2_ emissions per unit GDP; however, its impact decreased gradually with time. Alternatively, the regression coefficients for the shares of petroleum were almost all negative: 0.772, −0.652, −0.893, −0.751, and −0.603, respectively, with only the coefficient in 2000 being positive. Therefore, it was revealed that the correlation between the share of petroleum and CO_2_ emissions per unit GDP shifted from positive to negative over the analysis period, where more recent increases in petroleum’s share after 2003 led to a decline in CO_2_ emissions per unit GDP. The regression coefficients for the shares of natural gas were all negative: −0.168, −0.210, −0.337, −0.437, and −0.439, respectively; although, the absolute values of these coefficients demonstrated an increasing trend. These findings indicate that the share of natural gas had a negative correlation with CO_2_ emissions per unit GDP, where continuous increases in the shares of natural gas led to a decline in CO_2_ emissions per unit of GDP, with its impact gradually increasing with time. Lastly, the regression coefficients for the shares of non-fossil fuel energy sources were also all negative: −0.985, −0.501, −0.649, −0.693, and −0.581, for the years 2000, 2005, 2010, 2015, and 2019, respectively, showing that an increase in the share of non-fossil fuel-based energy sources leads to a decline in CO_2_ emissions per unit GDP; although, its impacts fluctuated over time.

The spatial lag terms in the SEMLD for 2005, 2010, 2015, and 2019 were significant, indicating the importance of the spatial spillover effect of CO_2_ emissions per unit GDP in each province. Moreover, the spatial lag terms in this model were 0.825, 0.811, 0.654, and 0.874, respectively, all of which were significant at the *α* = 0.10 level. These findings suggest that on average, for every 1% increase in CO_2_ emissions per unit GDP within a given province, correlated CO_2_ emissions per unit GDP increased by 0.825%, 0.811%, 0.654%, and 0.874%, respectively.

## 4. Discussion

### 4.1. Transformation of Energy Structure in China

(1)The spatial distribution of the share of energy resources varies by energy source type. Coal use is ubiquitous, but peaks in central and western China, whereas petroleum use peaked in southern China, and natural gas in western and southwestern China. Alternatively, the epicenter of non-fossil fuel energy use was mainly situated in southwestern and southern China.(2)The change trends in the share of energy sources varied by type. The share of coal first increased, then subsequently decreased; whereas the share of petroleum decreased, increased, then decreased again. The share of natural gas showed an upward trend, while the same was also observed for that of non-fossil fuel energy resources.(3)The pace of change in the share of energy sources varied by type as well. Non-fossil energy consumption changed at the fastest pace, especially more recently; whereas petroleum and coal consumption changed more gradually. Alternatively, natural gas consumption changed at a moderate pace. Overall, there was relatively little change observed throughout China’s energy infrastructure due to the low base effect.

### 4.2. Relationship between Energy Structure and Carbon Emissions

(1)Despite the different trends of change in China’s total, per capita, and per unit GDP CO_2_ emissions, energy infrastructure essentially had the same impact on all three carbon emissions indicators. Coal has a net positive impact on the indicators, while the impacts of petroleum were positive at first, turning negative thereafter. Meanwhile, both natural gas and non-fossil fuel energy resources had a primarily negative impact on these indicators. The main reason behind these findings is that the amount of CO_2_ produced varies by energy source type per unit energy produced. Based on the ton of coal equivalent (29,307.6 × 10³ kJ), carbon emissions from coal, petroleum, natural gas, and non-fossil energy were 2.50, 2.07, 1.61, and 0 carbon dioxide tons, respectively (without considering indirect carbon emissions caused by production processes). Hence, under the same conditions, provinces with higher shares of coal contributed more to CO_2_ emissions; whereas those with higher shares of natural gas and non-fossil fuel energy sources contributed less so.(2)The statistical results of the three models, namely, the SLM (8), SEM (9), and SEMLD, were all significant, thus indicating that energy infrastructure is correlated with total, per capita, and per unit GDP CO_2_ emissions. Nevertheless, since the SEMLD had the greatest LogL value, in addition to the lowest AIC and SC values, it demonstrated the best goodness of fit among the three. Therefore, the results obtained using the SEMLD were chosen as the basis for correlation analysis in the present study.(3)According to the SEMLD results, the impacts of energy infrastructure on carbon emissions varied widely over the analysis period, while varying effects on different carbon emissions indicators were observed as well. With respect to the impacts of energy infrastructure on total CO_2_ emissions, the strength of coal’s effects declined gradually over time, but remained the greatest of all energy source types. Alternatively, the impacts of petroleum on this indicator fluctuated, with an overall decreasing trend. Further, the impacts of natural gas on this indicator increased gradually with time, while those of non-fossil fuel energy sources fluctuated, with an overall upward trend. Regarding the impacts of energy structure on per capita CO_2_ emissions, coal fluctuated across a generally decreasing trend, while petroleum on this indicator declined at first, and increased thereafter. Conversely, the impacts of natural gas on this indicator rose gradually over time; whereas those of non-fossil fuel energy sources exhibited a continuously increasing trend, and were the greatest among all energy source types. Regarding the impact of energy structure on CO_2_ emissions per unit GDP, coal declined gradually over time, but was generally the greatest among all energy source types, while the impacts of petroleum fluctuated at first, before declining. Natural gas had an increasing impact on this indicator over time, and that of non-fossil fuel energy sources fluctuated across a generally downward trend. In summary, coal had the greatest impact on total and per unit GDP CO_2_ emissions, while non-fossil fuel energy sources and natural gas had the greatest impacts on per capita CO_2_ emissions.(4)Based on the spatial lag terms in the SEMLD, carbon emissions in a particular province were influenced not only by the energy infrastructure within, but also by the energy structure of neighboring provinces as well. On average, for every 1% increase in carbon emissions around a particular province, carbon emissions in the province increased by 0.5% to 0.9%. Hence, carbon emissions in neighboring provinces have a substantial impact on carbon emissions in a particular province.(5)According to the spatial error terms in the SEMLD, carbon emissions are not only significantly influenced by energy structure, but by other factors as well.

### 4.3. Policy Implications

After analyzing the characteristics of energy structure in China, and investigating the correlated impacts on carbon emissions, this study has put forth several suggestions for reducing carbon emissions within the energy sector, as listed below:

(1)Energy transitioning is paramount to achieving carbon reduction; thus, China must set a clear direction toward energy transitioning in the future. For instance, natural gas and non-fossil fuel energy resources should be prioritized, followed by petroleum, while coal consumption should be phased out.(2)During the energy transition process, interactions between provinces should not be overlooked, as it remains necessary to strengthen energy exchanges between provinces for improving comprehensive energy efficiency. With the central and western regions’ rich hydro- and solar energy resources, as well as the coastal region’s abundant wind energy resources, electricity can be transmitted and distributed via the grid between regions to bolster the proportion of non-fossil fuel energy consumption in all neighboring provinces as well. In addition, international exchanges should also be strengthened [44].(3)Since carbon emission reductions are a complicated target, a series of effective measures in addition to energy transitioning can also be implemented, such as improving energy efficiency, and increasing carbon sink capacity.

### 4.4. Limitations and Future Directions

There are several limitations with the present research, as listed below:(1)This study did not explain the mechanism(s) behind the impacts of energy structure on carbon emissions, especially with respect to the variable strengths of impact observed. While the impacts of energy resources varied by type, the reasons behind such variations remained unexplored.(2)Although this study investigated the impact of China’s overall energy structure on carbon emissions based on provincial panel data from 1997 to 2019, these varied by province. Owing to the significant presence of spatial autocorrelation in carbon emissions, geographically weighted regression model analyses are essential, the results of which will be discussed in a forthcoming paper.(3)The present study confirmed the significant negative impacts of natural gas and petroleum on the reduction in carbon emissions; however, since both are high-carbon energy sources, there should be a boundary for such negative impacts, which has yet to be determined.

Accordingly, this study proposes a few essential directions for future research:(1)By analyzing energy flow under the whole life cycle, and energy exchanges between provinces, specific details of energy consumption can be identified to explore the specific factors driving the impacts of energy structure on carbon emissions.(2)The impacts of energy structure on carbon emissions in different provinces can be studied using geographically weighted regression models, so as to propose more targeted strategies for energy transitioning in each region.(3)Strengthen research on the cost of energy transition; the current renewable energy (zero carbon emissions) reserves are large, but are limited by technology and cost constraints, and cannot be quickly large-scale promotion, which is also the direction of future research [44,45].(4)Strengthen the study of sectoral carbon emissions, combined with the analysis of sector-specific energy use and carbon emissions in the “China Energy Sector Carbon Neutral Roadmap” released by the International Energy Agency in September 2020; there is also a need to explore the relationship between energy structure and carbon emissions from different sectors in the future, especially key carbon-emitting sectors and enterprises, such as cement, iron and steel, chemical industry, transportation sector [46,47], etc.

## 5. Conclusions

(1)China’s energy structure primarily consists of coal, petroleum, natural gas, nuclear, hydro, wind, and solar power. Notably, coal accounts for the highest proportion of the country’s energy infrastructure; however, its share has generally declined with time. Alternatively, petroleum constitutes the second-highest proportion of the country’s energy structure, but its share has also generally decreased with time. Non-fossil fuel energy resources maintain the third-largest share of energy infrastructure, and exhibit a rapidly increasing trend. Lastly, natural gas accounts for the lowest proportion of the country’s energy infrastructure; although, its share has been increasing steadily over time.(2)Since 1997, there has been a continuous increase in China’s total and per capita CO_2_ emissions, albeit at a slower pace in recent years. The epicenters of peak emissions have shifted from the eastern region to the central and western regions. Conversely, CO_2_ emissions per unit GDP have exhibited a continuous decreasing trend at a slightly slower pace, while its peak emissions have shifted in identical geographic directions.(3)According to the results of the spatial autocorrelation analyses, there were significant spatial relationships between China’s total, per capita, and per unit GDP CO_2_ emissions, with the high-high cluster being dominant.(4)The results of the spatial regression model showed that in China’s energy infrastructure, coal has a positive impact on carbon emissions; whereas natural gas and non-fossil fuel energy sources have a negative impact. The impact of petroleum on carbon emissions, however, turned from positive to negative over the analysis period. Simultaneously, the impact of energy sources on different carbon emissions indicators varied by energy source type, where coal had the greatest impact on total and per capita CO_2_ emissions, while non-fossil fuel energy sources and natural gas maintained the greatest impact on CO_2_ emissions per unit GDP.

## Figures and Tables

**Figure 1 ijerph-19-05850-f001:**
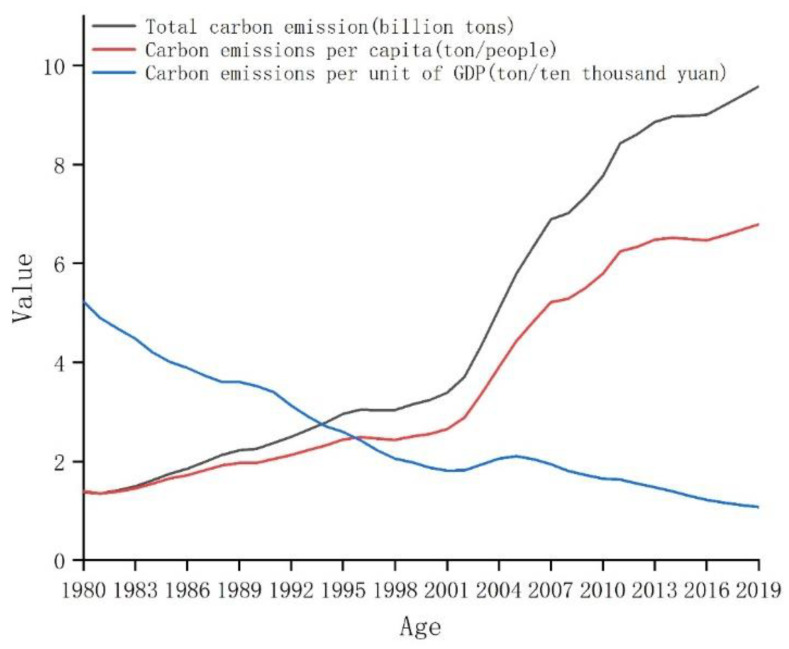
Total, per capita, and per unit GDP CO_2_ emissions in China since 1980 (source: China Energy Statistical Yearbook, China Statistical Yearbook, and China Emission Accounts and Datasets).

**Figure 2 ijerph-19-05850-f002:**
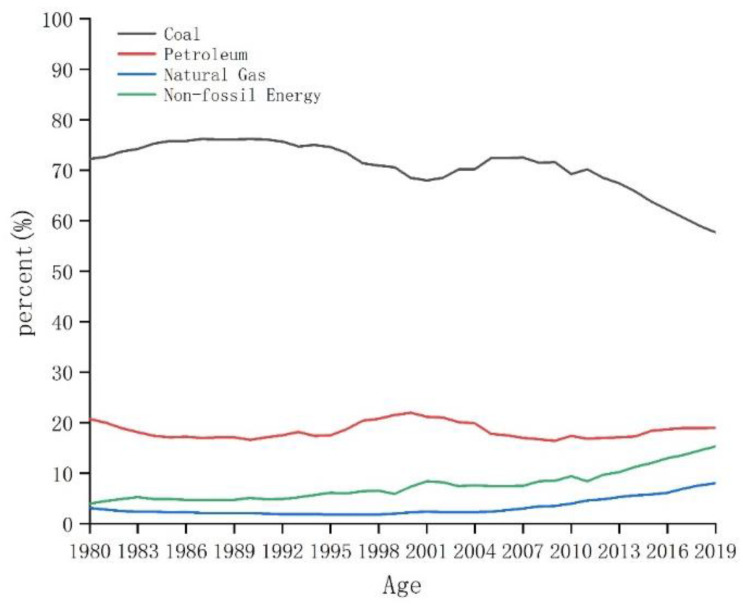
Energy structure of China since 1980 (source: China Energy Statistical Yearbook).

**Figure 3 ijerph-19-05850-f003:**
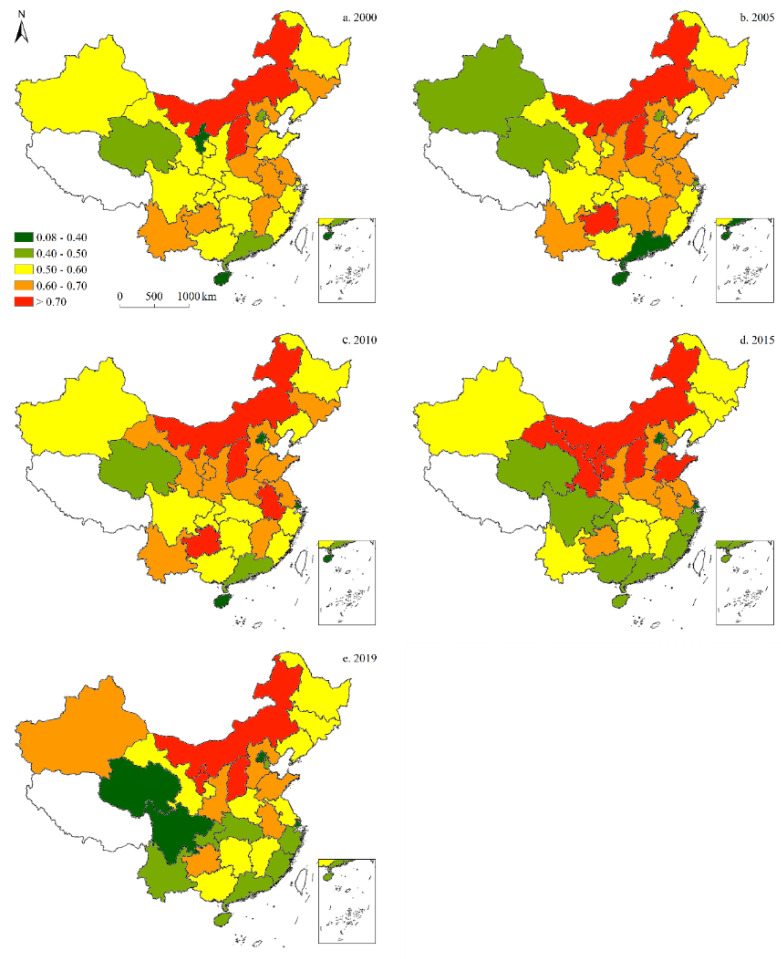
Spatiotemporal distribution of provincial-level coal consumption in China (five-time sections).

**Figure 4 ijerph-19-05850-f004:**
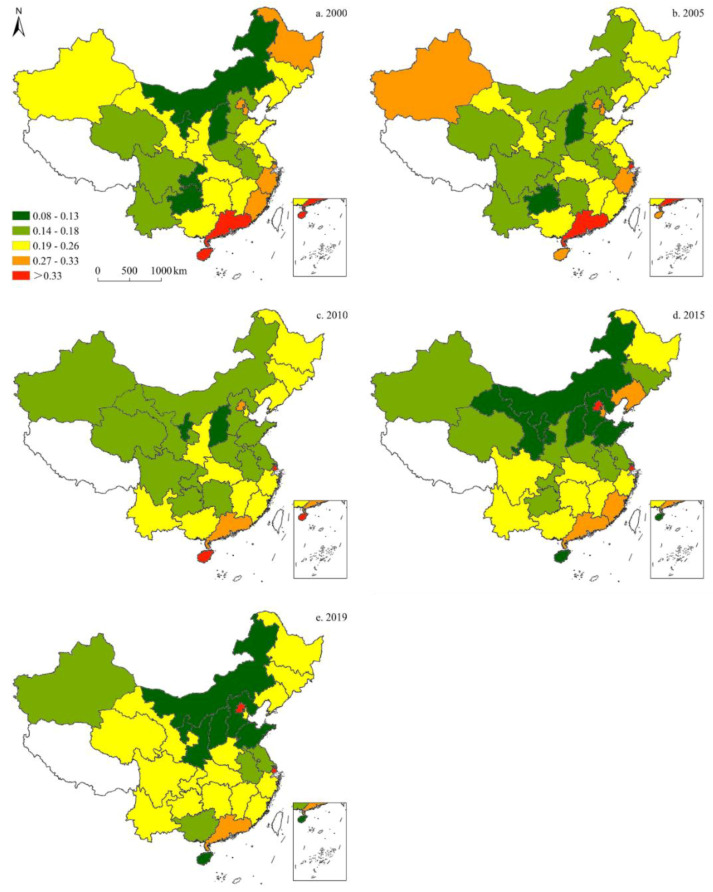
Spatiotemporal distribution of provincial-level petroleum consumption in China (five-time sections).

**Figure 5 ijerph-19-05850-f005:**
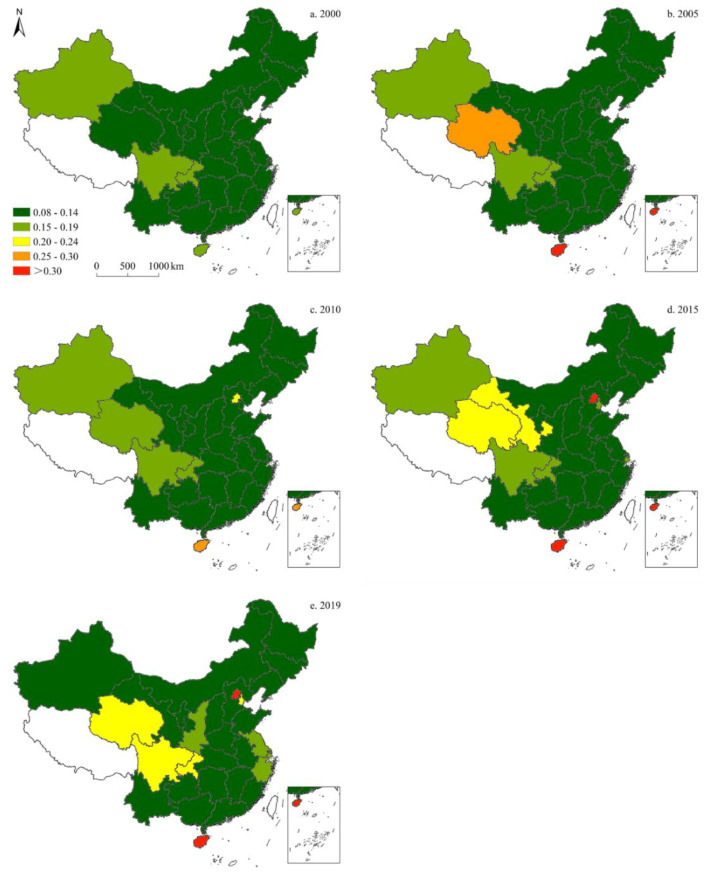
Spatiotemporal distribution of provincial-level natural gas consumption in China (five-time sections).

**Figure 6 ijerph-19-05850-f006:**
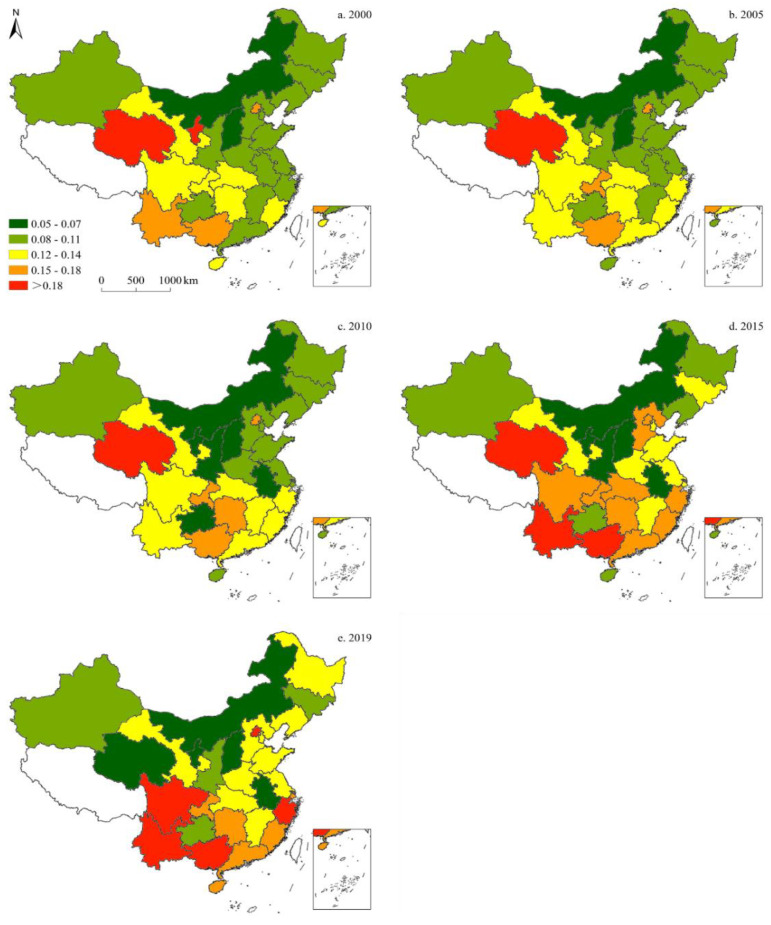
Spatiotemporal distribution of provincial-level non-fossil fuel energy consumption in China (five-time sections).

**Figure 7 ijerph-19-05850-f007:**
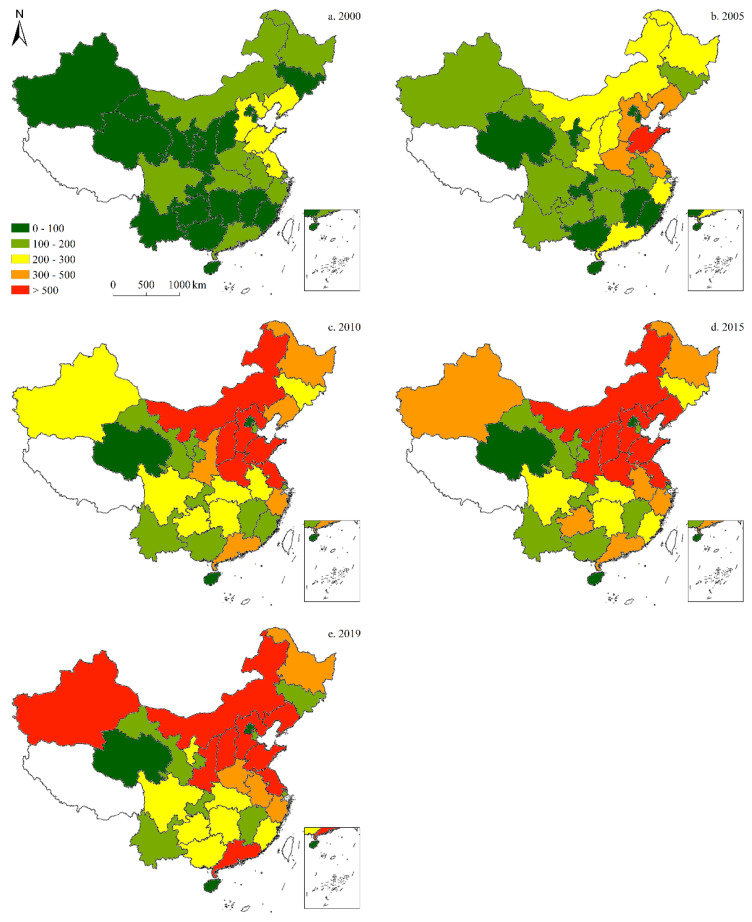
Spatiotemporal distribution of provincial-level total CO_2_ emissions (in million tons) in China since 2000.

**Figure 8 ijerph-19-05850-f008:**
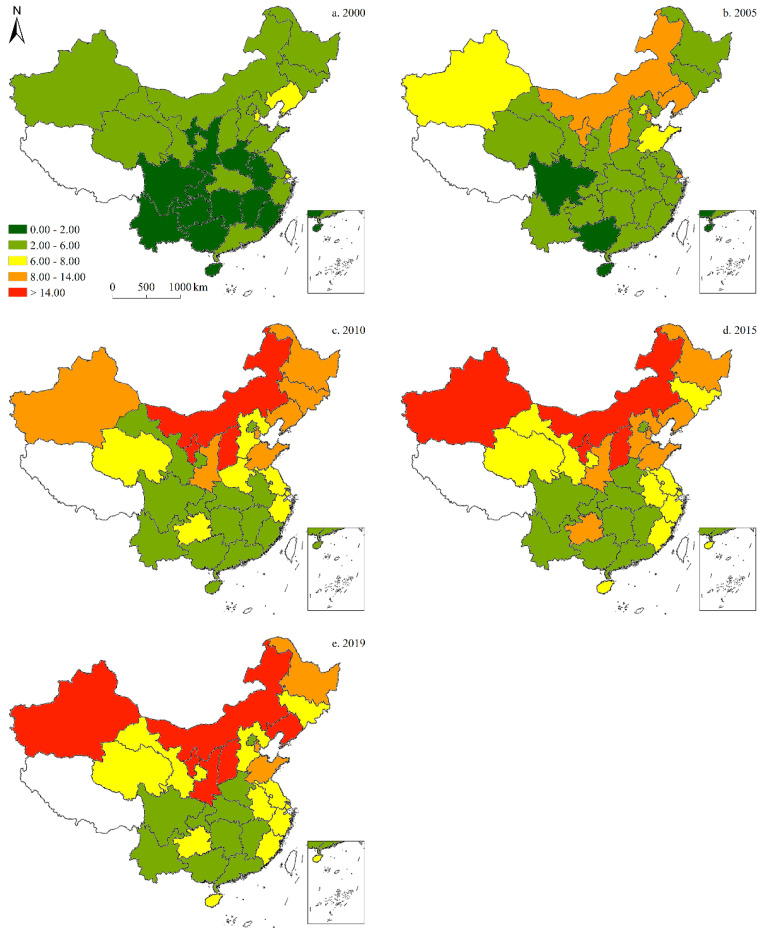
Spatiotemporal distribution of provincial-level CO_2_ emissions per capita (in tons·person^−1^) in China (five-time sections).

**Figure 9 ijerph-19-05850-f009:**
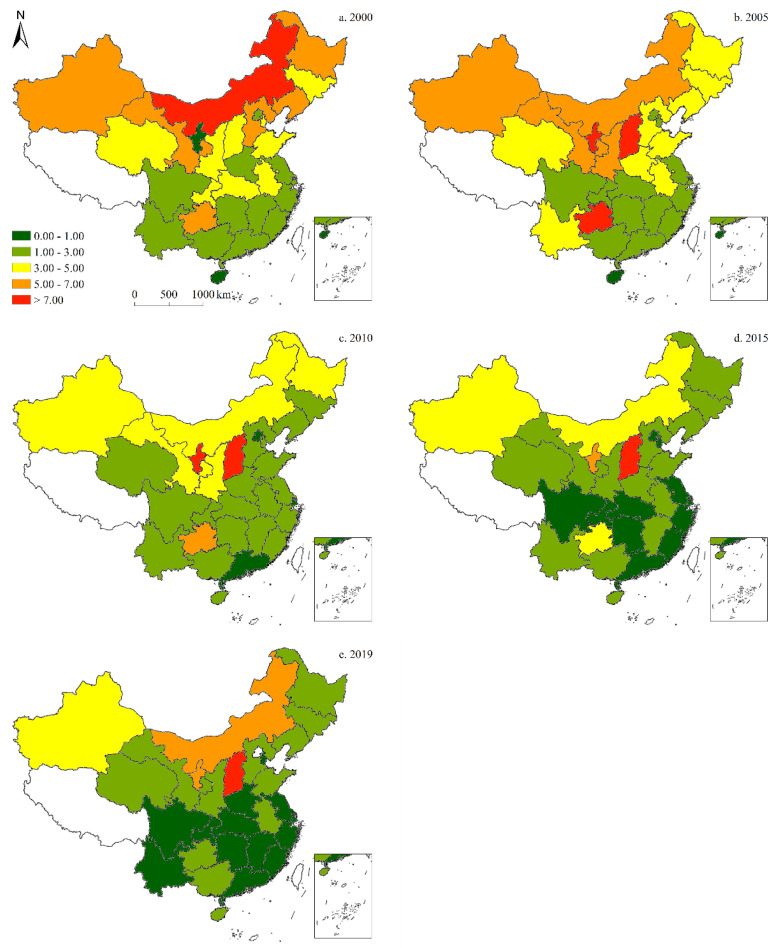
Spatiotemporal distribution of provincial-level CO_2_ emissions per unit of GDP (in tons· CNY 10,000^−1^) in China (five-time sections).

**Figure 10 ijerph-19-05850-f010:**
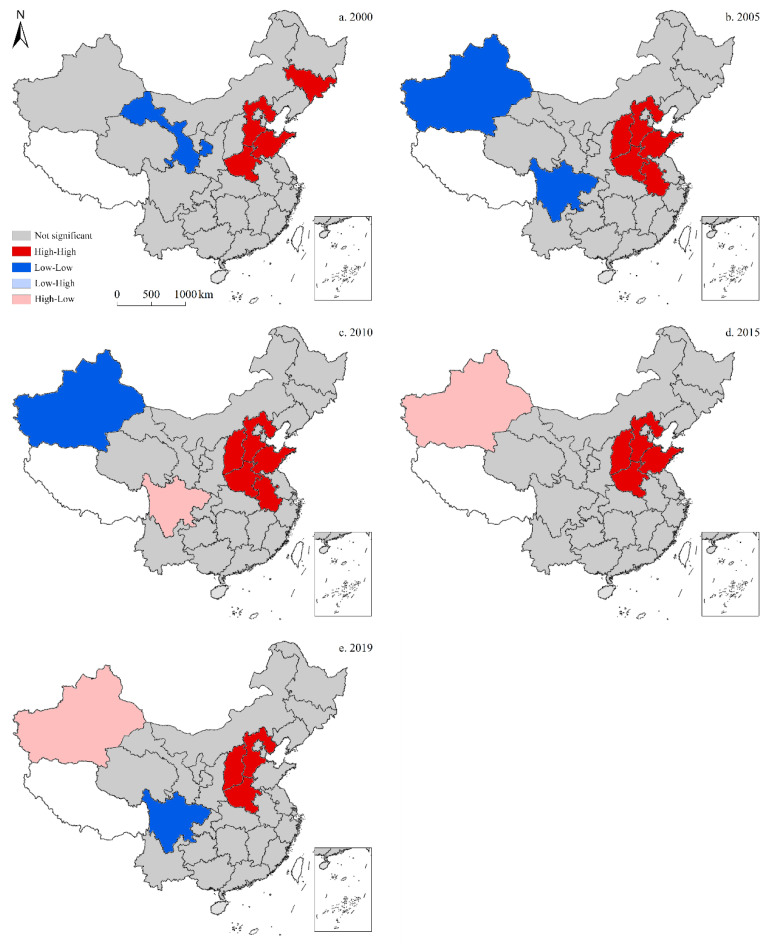
Local indicators of spatial association (LISA) map for the provincial-level spatiotemporal distribution of total CO_2_ emissions in China (five-time sections).

**Figure 11 ijerph-19-05850-f011:**
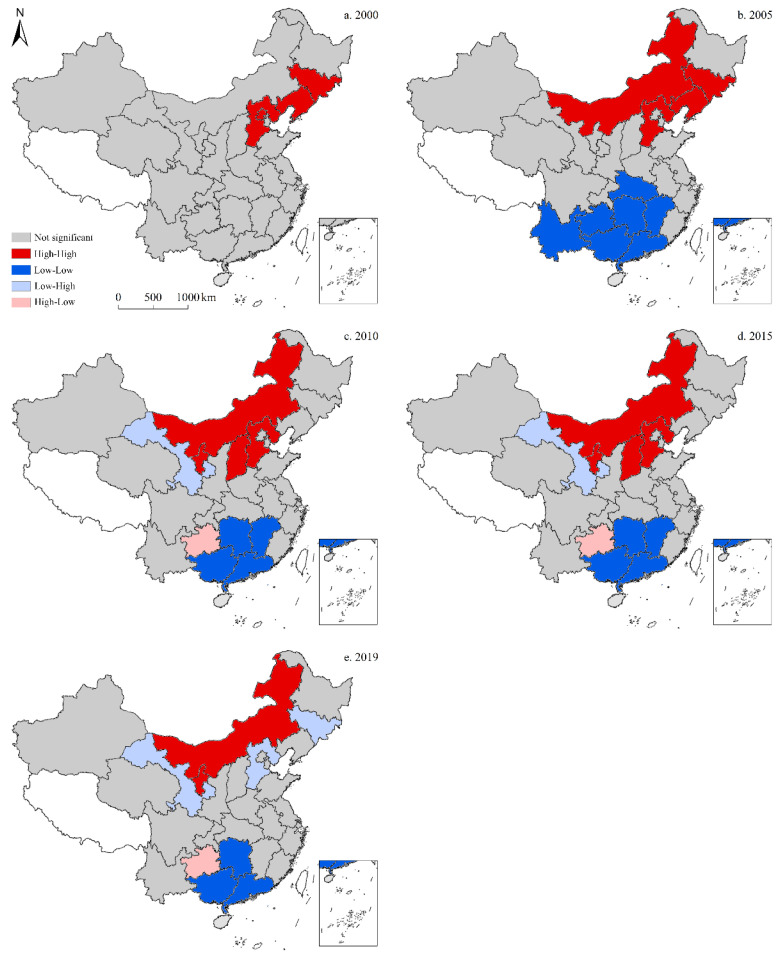
LISA map for the provincial-level spatiotemporal distribution of CO_2_ emissions per capita in China (five-time sections).

**Figure 12 ijerph-19-05850-f012:**
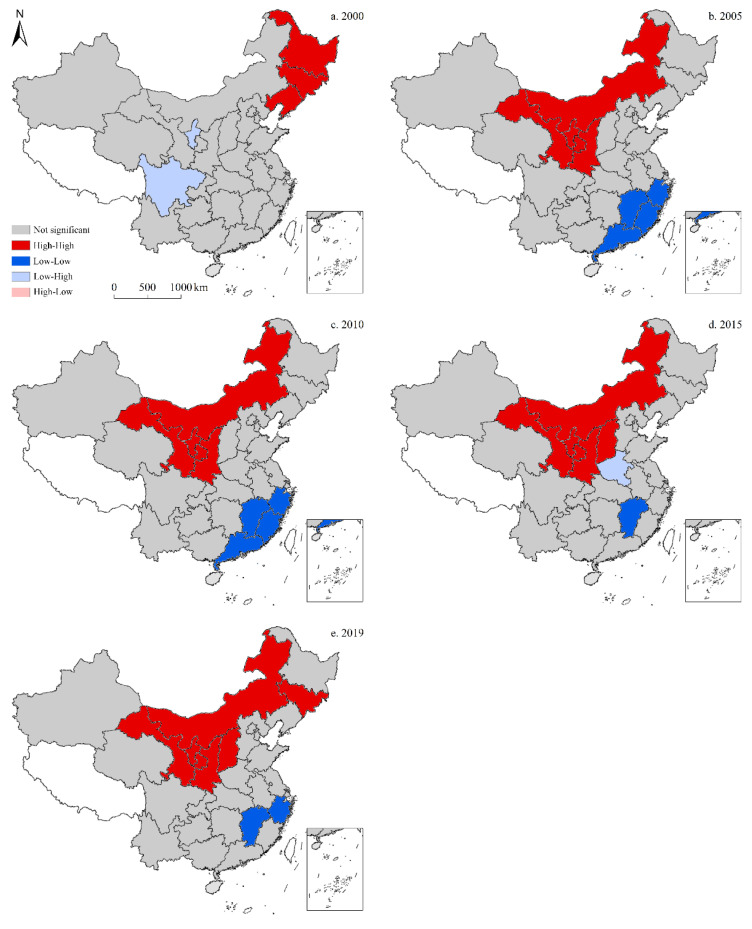
LISA map for the provincial-level spatiotemporal distribution of CO_2_ emissions per unit GDP in China (five-time sections).

**Table 1 ijerph-19-05850-t001:** Spatial autocorrelation indicators of carbon emissions in China from 2000 to 2019.

Year	CO_2_ Emissions per Unit of GDP (Tons per Ten Thousand CNY)			CO_2_ Emissions per Capita (Tons per Person)			Total CO_2_ Emissions (Million Tons)		
	Moran’s *I*	*z*-Score	*p*-Value	Moran’s *I*	*z*-Score	*p*-Value	Moran’s *I*	*z*-Score	*p*-Value
2000	−0.019	0.212	0.832	0.247	3.919	0.000	0.054	1.305	0.192
2005	0.130	2.183	0.029	0.400	5.777	0.000	0.075	1.531	0.126
2010	0.116	2.001	0.045	0.270	4.067	0.000	0.009	0.581	0.561
2015	0.076	1.481	0.139	0.166	2.687	0.007	−0.018	0.221	0.825
2019	0.111	1.934	0.053	0.146	2.421	0.015	−0.030	0.059	0.953

**Table 2 ijerph-19-05850-t002:** Spatial regression results for total CO_2_ emissions in China (five-time sections).

	2000				2005				2010			
	OLS	SLM	SEM	SEMLD	OLS	SLM	SEM	SEMLD	OLS	SLM	SEM	SEMLD
R^2^	0.742	0.828	0.787	0.871	0.450	0.454	0.452	0.567	0.410	0.435	0.410	0.578
LogL	−37.901	−32.554	−36.242	−26.730	−32.428	−32.336	−32.395	−30.254	−33.015	−32.543	−33.014	−28.608
AIC	85.801	77.108	82.485	65.460	74.855	76.672	74.791	72.507	76.031	77.086	76.029	69.215
SC	92.807	85.515	89.491	73.663	81.692	84.876	81.627	80.711	82.867	85.289	82.865	77.419
	**2015**				**2019**							
R^2^	0.501	0.517	0.501	0.671	0.464	0.468	0.464	0.520				
LogL	−33.086	−32.739	−33.083	−29.981	−35.923	−35.839	−35.920	−32.893				
AIC	76.171	77.479	76.166	71.962	81.846	83.678	81.840	77.785				
SC	83.008	85.682	83.003	80.166	88.682	91.881	88.677	85.989				

**Table 3 ijerph-19-05850-t003:** Spatial regression results for total CO_2_ emissions in China (five-time sections), based on the SEMLD model.

	2000	2005	2010	2015	2019
Constant (*γ*)	−6.180 **	−0.432	−16.915	−5.174	2.575
	(3.052)	(19.148)	(20.135)	(4.903)	(7.953)
Coal (*α*)	1.523 ***	0.850	0.652	1.284 ***	0.723
	(0.299)	(2.149)	(2.165)	(0.477)	(0.686)
Petroleum (*α*)	0.537 ***	0.239	0.160	−0.114 **	−0.288
	(0.167)	(0.823)	(0.834)	(0.053)	(0.434)
Natural gas (*α*)	−0.085	−0.065	−0.074	−0.126	−0.190
	(0.312)	(0.616)	(0.711)	(0.352)	(0.632)
Non-fossil energy (*α*)	−0.619 **	−0.500	−0.019	−0.011	−0.196
	(0.249)	(0.518)	(0.505)	(0.312)	(0.390)
Spatial lag term (*β*_1_)	0.506 ***	0.628 ***	0.744 ***	0.776 ***	0.504 **
	(0.154)	(0.221)	(0.221)	(0.205)	(0.242)
Spatial error term (*β*_2_)	−0.915 ***	−0.736 ***	−0.678 ***	−0.952 ***	−0.382
	(0.238)	(0.258)	(0.263)	(0.232)	(0.276)
R^2^	0.871	0.567	0.578	0.671	0.520
LogL	−26.730	−30.254	−28.608	−29.981	−32.893
AIC	65.460	72.507	69.215	71.962	77.785
SC	73.663	80.711	77.419	80.166	85.989
*N*	30	30	30	30	30

The present study constructed spatial matrices using queen contiguity. *** *p* ≤ 0.01, ** *p* ≤ 0.05. Values in parentheses denote standard deviations.

**Table 4 ijerph-19-05850-t004:** Spatial regression results for per CO_2_ emissions in China (five-time sections).

	2000				2005				2010			
	OLS	SLM	SEM	SEMLD	OLS	SLM	SEM	SEMLD	OLS	SLM	SEM	SEMLD
R^2^	0.703	0.779	0.860	0.907	0.638	0.736	0.709	0.802	0.605	0.695	0.643	0.809
LogL	−21.451	−18.137	−13.992	−10.302	−19.119	−15.374	−17.572	−12.016	−20.361	−17.230	−19.470	−12.873
AIC	52.902	48.274	37.983	34.605	50.238	44.749	47.144	38.032	52.722	48.459	50.940	39.746
SC	59.739	56.478	44.820	44.176	58.442	54.320	55.348	47.603	60.926	58.030	59.143	49.317
	**2015**				**2019**							
R^2^	0.772	0.815	0.806	0.817	0.774	0.780	0.774	0.792				
LogL	−17.740	−15.079	−16.216	−14.568	−20.479	−20.146	−20.471	−19.630				
AIC	47.481	44.158	44.431	43.137	52.958	54.292	52.942	53.260				
SC	55.685	53.730	52.635	52.708	61.162	63.863	61.146	62.831				

**Table 5 ijerph-19-05850-t005:** Spatial regression results for per capita CO_2_ emissions in China (five-time sections), based on the SEMLD model.

	2000	2005	2010	2015	2019
Constant (*γ*)	−3.636 *	17.069 *	19.231 *	0.970	8.919 **
	(2.054)	(10.127)	(10.354)	(3.104)	(4.465)
Coal (*α*)	0.811 ***	0.712	0.656	0.856 ***	0.110
	(0.250)	(1.164)	(1.166)	(0.324)	(0.409)
Petroleum (*α*)	0.630 ***	0.367	−0.569	−0.058 *	−0.619 **
	(0.107)	(0.451)	(0.458)	(0.032)	(0.263)
Natural gas (*α*)	−0.428 **	−0.348	−0.530	−0.623	−0.757
	(0.206)	(0.359)	(0.372)	(0.307)	(0.375)
Non-fossil energy (*α*)	−0.619 ***	−1.035 ***	−0.926 ***	−0.760 ***	−0.798 ***
	(0.199)	(0.308)	(0.255)	(0.186)	(0.252)
Spatial lag term (*β*_1_)	0.651 ***	0.693 ***	0.681 ***	0.377 **	0.251 *
	(0.113)	(0.133)	(0.104)	(0.153)	(0.152)
Spatial error term (*β*_2_)	−0.610 **	−0.709 ***	−0.965 ***	0.077	−0.344
	(0.268)	(0.260)	(0.230)	(0.255)	(0.276)
R^2^	0.907	0.802	0.809	0.817	0.792
LogL	−10.302	−12.016	−12.873	−14.568	−19.630
AIC	34.605	38.032	39.746	43.137	53.260
SC	44.176	47.603	49.317	52.708	62.831
*N*	30	30	30	30	30

The present study constructed spatial matrices using queen contiguity. *** *p* ≤ 0.01, ** *p* ≤ 0.05, * *p* ≤ 0.1. Values in parentheses denote standard deviations.

**Table 6 ijerph-19-05850-t006:** Spatial regression results for CO_2_ emissions per unit GDP in China (five-time sections).

	2000				2005				2010			
	OLS	SLM	SEM	SEMLD	OLS	SLM	SEM	SEMLD	OLS	SLM	SEM	SEMLD
R^2^	0.805	0.811	0.810	0.812	0.624	0.751	0.758	0.772	0.659	0.804	0.846	0.812
LogL	−30.067	−29.680	−29.909	−29.660	−20.491	−15.889	−17.345	−13.569	−21.418	−15.074	−13.930	−12.797
AIC	70.134	71.359	69.817	71.320	50.983	43.779	44.689	39.138	52.836	42.147	37.860	37.595
SC	76.970	79.563	76.654	79.524	57.819	51.983	51.526	47.342	59.673	50.351	44.696	45.799
	**2015**				**2019**							
R^2^	0.690	0.790	0.785	0.806	0.743	0.837	0.842	0.839				
LogL	−26.335	−21.821	−22.906	−19.859	−28.344	−22.709	−23.242	−21.556				
AIC	62.669	55.643	55.813	51.719	66.688	57.417	56.483	55.112				
SC	69.506	63.847	62.649	59.923	73.524	65.621	63.320	63.316				

**Table 7 ijerph-19-05850-t007:** Spatial regression results for CO_2_ emissions per unit GDP in China (five-time sections), based on the SEMLD model.

	2000	2005	2010	2015	2019
Constant (*γ*)	−0.269	5.040	10.387	−4.082	1.296
	(4.017)	(10.722)	(9.787)	(3.685)	(4.544)
Coal (*α*)	1.181 ***	1.175	1.133	1.129 ***	0.607
	(0.456)	(1.221)	(1.095)	(0.412)	(0.420)
Petroleum (*α*)	0.772 ***	−0.652	−0.893 **	−0.751	−0.603 **
	(0.257)	(0.452)	(0.432)	(0.038)	(0.282)
Natural gas (*α*)	−0.168	−0.210	−0.337	−0.437	−0.439
	(0.467)	(0.419)	(0.412)	(0.384)	(0.413)
Non-fossil energy (*α*)	−0.985 ***	−0.501 *	−0.649 ***	−0.693 ***	−0.581 **
	(0.361)	(0.285)	(0.236)	(0.206)	(0.249)
Spatial lag term (*β*_1_)	0.217	0.825 ***	0.811 ***	0.654 ***	0.874 ***
	(0.301)	(0.168)	(0.166)	(0.203)	(0.140)
Spatial error term (*β*_2_)	−0.190	−0.330	−0.005	0.282	1.17487
	(0.273)	(0.276)	(0.262)	(0.229)	(0.260)
R^2^	0.812	0.772	0.812	0.806	0.839
LogL	−29.660	−13.569	−12.797	−19.859	−21.556
AIC	71.320	39.138	37.595	51.719	55.112
SC	79.524	47.342	45.799	59.923	63.316
*N*	30	30	30	30	30

The present study constructed spatial matrices using queen contiguity. *** *p* ≤ 0.01, ** *p* ≤ 0.05, * *p* ≤ 0.1. Values in parentheses denote standard deviations.

## Data Availability

China Statistical Yearbook from China National Bureau of Statistics (http://www.stats.gov.cn/, accessed on 1 May 2021); China Energy Statistical Yearbook from China National Energy Administration (http://www.nea.gov.cn/, accessed on 1 May 2021); Emission data from China Carbon Accounting Database (https://www.ceads.net.cn/, accessed on 1 May 2021); 2006 IPCC Guidelines for National Greenhouse Gas Inventories (https://www.ipcc.ch/, accessed on 1 May 2021). All above data are publicly available and there is no copyright dispute.

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
