# Peer review of "Impacts of Energy Structure on Carbon Emissions in China, 1997–2019"

_ijerph, 2022, doi:10.3390/ijerph19105850_

Round 1

Reviewer 1 Report

All suggestions were considered. According to my opinion, the manuscript has now improved and is ready for publication

Reviewer 2 Report

The authors change well and it can be accepted.

This manuscript is a resubmission of an earlier submission. The following is a list of the peer review reports and author responses from that submission.

Round 1

Reviewer 1 Report

The authors proposed that Based on energy consumption and carbon emis-sions data from 30 provincial-level administrative regions in China (excluding Tibet, Hong Kong, Taiwan, and Macau due to the lack of data), the study here investigated the shares of coal, petro-leum, natural gas, and non-fossil energy sources (i.e., hydropower, nuclear power, wind power, solar power, and biomass power), as they relate to total, per capita, and per unit GDP CO2 emissions via spatial regression. The results are reasonable and the manuscript is well-organized. However, there are some minor problems about the manuscript, therefore I suggest this paper for "minor revision" in the journal of International Journal of Environmental Research and Public Health.

1.In my opinion this manuscript abstract should shorten and highlight the distinguishing feature of the article. The format of the article, especially the recently 3 years reference.

2.I think the graphic abstract should be added and should shorten the abstract.

3.The authors may discuss deep learning for the other field and expand the readership. Therefore, the introduction of recent progress energy structure on carbon emissions and energy storage should be cited. 

4.The figure and table caption should be more informative.

5.What’s the innvotion of this article? The author should explain.

6.I suggest the author tell the novelty of the article and suggest to compare with others.

In short, in its current form, the paper is not suitable for acceptance. The paper needs rewriting, by addressing the above-mentioned comments.

Author Response

I would like to thank the academic editor for his comments on my manuscript, which will be of great significance to improve this thesis and future research. I fully accept your comments, and now I am going to explain the changes one by one.

  1. Six new papers have been added, and the data and background of the study are all from the last 3 years, for example, China's carbon neutrality and carbon peak targets were proposed in September 2020.
  2. The abstract has been revised to refine the statements.
  3. The introduction of energy mix has been added.
  4. The title and position of the graphs have been revised.
  5. The motivation of this article has been stated in the text, and this paper has further elaborated on this basis, mainly to guide the energy mix transition, and to provide some reference for China to achieve peak carbon by 2030 and carbon neutrality by 2060.
  6. The novelty of the article has been appropriately added in the abstract and conclusion.

Finally, thank you again for your valuable advice.

Reviewer 2 Report

Based on energy consumption and carbon emissions data from 30 provincial-level administrative regions in China, the study here investigated the shares of coal, petroleum, natural gas, and non-fossil energy source, as they relate to total, per capita, and per unit GDP CO2 emissions via spatial regression.

The subject matter is relevant with particular reference to energy transition goals. Therefore, it is a topical issue. Moreover, the topic is addressed in an organic manner.

In my opinion, however, the article has two limitations: 1. The bibliographic analysis is not complete and needs to be integrated. I suggest reading and considering for references: AlKhars, M.; Miah, F.; Qudrat-Ullah, H.; Kayal, A. A Systematic Review of the Relationship Between Energy Consumption and Economic Growth in GCC Countries. Sustainability 2020, 12, 3845.  Nesticò, A. and Maselli, G. Declining discount rate estimate in the long-term economic evaluation of environmental projects. Journal of Environmental Accounting and Management 2020, Vol. 8, Issue 1, pp. 93-110. https://doi.org/10.5890/JEAM.2020.03.007; 2) In my opinion, the importance of the topic for decision making should also be highlighted in the conclusions of the work. Moreover, I suggest to highlight the profiles of originality of the paper in the conclusions.

Author Response

Dear reviewers, thank you very much for your comments, they have helped me a lot. I fully accept your comments and have made changes in accordance with your comments as follows:
1. The first paper proposes the same policy to increase the proportion of renewable energy as this paper, in the "Policy Implications" and "Limitations and Future Directions" sections of the paper. "The second paper suggests that "The decision-making processes regarding projects with long-term environmental implications are strongly influenced by the estimate of the Social Discount Rate (SDR)", which is very useful in the "Limitations and Future Directions" section of the study. It is useful to consider the long-term input costs in the energy transition, see literature 44-45.
2. New countermeasures have been added to the study, and later the participants of the energy transition, especially the government and the companies, will be added, which has been discussed using game theory.
Finally thank you very much for your valuable comments and thank you for your work!

Reviewer 3 Report

The manuscript is well written and organized. The argument is original and aligned with the scope of the journal. According to my opinion, it could be accepted for publication after minor improvements.

What can be said about means of transport and energy resources and their environmental impact. Refer for example to "Spreafico, C., & Russo, D. (2020). Exploiting the scientific literature for performing life cycle assessment about transportation. Sustainability, 12 (18), 7548." or "Velazquez, L., Munguia, N. E., Will, M., Zavala, A. G., Verdugo, S. P., Delakowitz, B., & Giannetti, B. (2015). Sustainable transportation strategies for decoupling road vehicle transport and carbon dioxide emissions . Management of Environmental Quality: An International Journal. "

Author Response

Dear Reviewer, Thank you very much for your comments, they have been very helpful. I fully accept your comments and have made changes according to your comments as follows:
Carbon emissions from the transportation sector are very large and have a significant impact on the energy mix, which I have highlighted in " Limitations and Future Directions, see Ref. 46-47.

Combined with the analysis of energy use and carbon emissions in transportation, cement, iron and steel, and chemical sectors in the "China Energy Sector Carbon Neutral Roadmap" released by the International Energy Agency in September 2020, I think the relationship between energy mix and carbon emissions needs to be explored in different sectors in the future, especially in the transportation sector.
Finally, thank you very much for the research direction of sectoral energy consumption and carbon emissions!
